# The Effect of Protein Supplementation versus Carbohydrate Supplementation on Muscle Damage Markers and Soreness Following a 15-km Road Race: A Double-Blind Randomized Controlled Trial

**DOI:** 10.3390/nu13030858

**Published:** 2021-03-05

**Authors:** Dominique S. M. ten Haaf, Martin A. Flipsen, Astrid M. H. Horstman, Hans Timmerman, Monique A. H. Steegers, Lisette C. P. G. M. de Groot, Thijs M. H. Eijsvogels, Maria T. E. Hopman

**Affiliations:** 1Department of Physiology, Radboud Institute for Health Sciences, Radboud University Medical Center, 6525 XZ Nijmegen, The Netherlands; dominique.tenhaaf@live.nl (D.S.M.t.H.); martin.flipsen@gmail.com (M.A.F.); Maria.Hopman@radboudumc.nl (M.T.E.H.); 2FrieslandCampina, 3811 LP Amersfoort, The Netherlands; Astrid.Horstman@rd.nestle.com; 3Department of Anesthesiology, Pain and Palliative Medicine, Radboud University Medical Center, 6525 XZ Nijmegen, The Netherlands; h.timmerman02@umcg.nl (H.T.); Monique.Steegers@radboudumc.nl (M.A.H.S.); 4Pain Center, Department of Anesthesiology, University Medical Center Groningen, 9713 GZ Groningen, The Netherlands; 5Department of Anesthesiology, Amsterdam University Medical Center Location VU, 1081 HV Amsterdam, The Netherlands; 6Division of Human Nutrition and Health, Wageningen University, 6708 PB Wageningen, The Netherlands; lisette.degroot@wur.nl

**Keywords:** milk protein, endurance exercise, muscle recovery, delayed onset of muscle soreness

## Abstract

We assessed whether a protein supplementation protocol could attenuate running-induced muscle soreness and other muscle damage markers compared to iso-caloric placebo supplementation. A double-blind randomized controlled trial was performed among 323 recreational runners (age 44 ± 11 years, 56% men) participating in a 15-km road race. Participants received milk protein or carbohydrate supplementation, for three consecutive days post-race. Habitual protein intake was assessed using 24 h recalls. Race characteristics were determined and muscle soreness was assessed with the Brief Pain Inventory at baseline and 1–3 days post-race. In a subgroup (*n* = 149) muscle soreness was measured with a strain gauge algometer and creatine kinase (CK) and lactate dehydrogenase (LDH) concentrations were measured. At baseline, no group-differences were observed for habitual protein intake (protein group: 79.9 ± 26.5 g/d versus placebo group: 82.0 ± 26.8 g/d, *p* = 0.49) and muscle soreness (protein: 0.45 ± 1.08 versus placebo: 0.44 ± 1.14, *p* = 0.96). Subjects completed the race with a running speed of 12 ± 2 km/h. With the Intention-to-Treat analysis no between-group differences were observed in reported muscle soreness. With the per-protocol analysis, however, the protein group reported higher muscle soreness 24 h post-race compared to the placebo group (2.96 ± 2.27 versus 2.46 ± 2.38, *p* = 0.039) and a lower pressure muscle pain threshold in the protein group compared to the placebo group (71.8 ± 30.0 N versus 83.9 ± 27.9 N, *p* = 0.019). No differences were found in concentrations of CK and LDH post-race between groups. Post-exercise protein supplementation is not more preferable than carbohydrate supplementation to reduce muscle soreness or other damage markers in recreational athletes with mostly a sufficient baseline protein intake running a 15-km road race.

## 1. Introduction

Regular physical activities results in a myriad of health benefits, such as a reduced risk for cardiovascular diseases, several cancers, and diabetes [1]. On the other hand, intense bouts of exercise training also result in micro injuries to contractile proteins, so-called muscle damage, as demonstrated by a post-exercise increase in muscle damage markers including delayed onset of muscle soreness [2]. Protein supplementation has been proposed as a promising strategy to attenuate exercise-induced muscle damage and soreness, since the intake of protein results in a positive muscle protein balance, which is critical for the remodeling of skeletal muscle [3,4,5]. 

Small-scale studies including few participants (*n* = 9 to *n* = 16, age ~22 years) reported positive effects of protein supplementation following high-intensity endurance exercise, on muscle damage markers including delayed onset of muscle soreness on subsequent days in healthy men with fitness levels ranging from recreational exercise to elite level [6,7,8,9]. However, other studies failed to replicate these findings in men and women with varying fitness levels and reported no difference in muscle damage markers including muscle soreness between the protein supplementation group and placebo group [10,11,12]. This discrepancy could be partly due to a difference in using a study protocol with or without multiple bouts of exercise, since the repeated bout effect could influence the findings. This repeated bout effect refers to the adaptation whereby a single bout of eccentric exercise protects against muscle damage from subsequent eccentric bouts [13]. A systematic review concluded that more attention should be given to the effects of different protein sources and timing of supplementation on muscle damage, soreness and recovery of muscle function and physical performance [4]. Both casein and whey protein in milk protein contain all essential amino acids required to effectively stimulate muscle protein synthesis [14,15]. Moreover, to support muscle protein synthesis an effective pattern of daily protein intake distribution is to provide at least 20 g of protein with each main meal with no more than 4–5 h between each meal [16,17,18,19] or after one-legged resistance type exercise [20]. An extra opportunity during the day for protein intake is to use at least 40 g of protein prior to sleep stimulate muscle protein synthesis until the next morning [21,22,23,24]. These thresholds are often not reached by adults [25,26]. 

On the other hand, it has been shown that carbohydrates inhibit protein breakdown as well due to an augmented insulin secretion [27,28]. It is unknown how these proposed effects of protein and carbohydrates intake on muscle protein balance translate in reducing exercise induced muscle soreness and muscle damage and whether only increasing the protein intake after exercise is superior in reducing exercise-induced muscle soreness and muscle damage compared to carbohydrates only. 

In the present randomized controlled trial we would like to compare an optimized protein supplementation protocol (20 g milk protein (20% whey protein, 80% casein) post-exercise and during breakfast and 40 g milk protein prior to sleep) to iso-caloric carbohydrate supplementation on exercise-induced muscle soreness and other markers of muscle damage. We hypothesized that recreational endurance athletes receiving protein supplementation would report lower muscle soreness and have lower concentrations of markers of muscle damage post-exercise compared to athletes receiving carbohydrate placebo supplementation. 

## 2. Methods

### 2.1. Participants

Participants were recruited between 31 October 2018 and 16 November 2018 via the Nijmegen Exercise Study database (study-ID: NL36743.091.11) [29] and social media. Interested men and women between the age of 25 and 65 years were included if they were registered for the Seven Hills Run (18 November 2018) (https://www.nnzevenheuvelenloop.nl/) (accessed on 15 November 2020) and were able to understand and perform the study procedures. The Seven Hills Run is an annual 15-km road race in Nijmegen, the Netherlands, and currently holds the men’s and women’s world record, but is mostly performed by recreational athletes, which were included in this study. Exclusion criteria for participation in the study were type I or type II diabetes mellitus, allergic or intolerant for milk ingredients, eggs, and soy beans, diagnosed with renal insufficiency or intestinal diseases that may influence the digestion and absorption of protein, use of statins and presence of muscle soreness or muscle complaints in daily life that are unrelated to exercise. All participants provided written consent prior to any experimental procedures. The study conformed to the principles of the Declaration of Helsinki and was approved by a local Medical Ethical committee, the Independent Review Board Nijmegen (Study-ID: NL67354.072.18). This trial was registered at www.trialregister.nl as NL7356.

### 2.2. Study Design

In this double-blind, placebo-controlled intervention study a total of 323 eligible participants out of 419 screened participants (Figure 1) were randomly allocated to either the protein supplement or the CHO placebo supplement group. An independent researcher randomized the study participants by means of computer-generated random numbers with a block size of 10 in a 1:1 ratio. 

Pre-race: Baseline (i.e., 1–3 days pre-race) muscle soreness was questioned using online versions of the Short-Form Brief Pain Inventory (BPI-SF) [30], including a Numeric Pain Rating Scale (NPRS) (scale: 0–10). Moreover, habitual dietary intake was determined using 24-h recalls. Information about demographics (i.e., age, sex, height, body weight), exercise training characteristics were collected from all participants.

During race: Due to the large number of participants in the race (±30,000), runners departed in nine separate “waves” between 13.00 and 14.00 CEST. Directly after finishing the race, participants received the first protein or placebo supplement to ingest immediately. Finish times were collected from all participants. Additionally, average heart rate during the race and exercise intensity was collected from participants that ran with a heart rate monitor.

Post-race: the participants received supplements to consume prior to sleep and during breakfast until three days post-race. Participants were instructed to fill out questionnaires about muscle soreness at one, two, and three days post-race. Moreover, study participants completed 24-h recalls about their dietary intake on the day of the race, and the first and second day post-race. 

Within a subgroup of 75 participants of the protein group and 74 participants of the placebo group, objective muscle soreness was measured with a strain gauge algometer (Wagner instruments, Force FX, Greenwhich, CT, USA) at 25–48 h post-race. Blood concentrations of creatine kinase (CK) and lactate dehydrogenase (LDH) were also assessed in this subgroup to determine muscle damage. An overview of the study procedures is presented in Figure 2.

### 2.3. Supplementation Protocol

Participants were instructed to consume either 1 serve of 250 mL dairy-based protein supplement (containing 20 g of milk protein; 80% casein, 20% whey protein) or 1 serve of 250 mL iso-caloric carbohydrate placebo drink directly after the race and during breakfast 1 day, 2 days and 3 days post-race, to ensure at least 20 g protein was consumed post-exercise and during breakfast. Moreover, prior to sleep the subjects were instructed to consume either 2 serves of the protein supplement (500 mL; containing 40 g of milk protein) or 2 serves of isocaloric placebo drink on the day of the race, and 1 day and 2 days post-race (500 mL), to ensure at least 40 g protein was consumed prior to sleep. On race day, participants received the supplements as a ready-to-drink product. The other supplements had to be consumed at home and were provided as powder; 1 serve of 3 spoons had to be mixed with 250 mL water for breakfast and for 2 serves, 6 spoons had to be mixed with 500 mL water prior to sleep. Shakers and spoons were added to the package with instructions (including pictures) on how to prepare the supplements, to ensure that the right mixture was prepared by the participants at home. One serve of the protein supplement contained 20 g of dairy milk protein, 0.6 g fat and 9.6 g carbohydrates, resulting in 123 kcal. One serve of the maltodextrin based placebo supplement contained 0 g protein, 0.6 g fat and 29.3 g carbohydrates, resulting in 123 kcal. Both protein and placebo supplements also contained 270 mg calcium, 660 mg potassium, 125 mg magnesium, 0.5 mg vitamin B6 and 1.65 µg vitamin D (FrieslandCampina, Amersfoort, The Netherlands). Two serves contained the double amounts. Both supplements were vanilla flavored to mask contents. Participants were asked to daily report their supplement intake. Compliance was determined per day by questioning the intake per moment (prior to sleep and with breakfast). Adverse events were documented throughout the study. 

### 2.4. Measurements 

#### 2.4.1. Demographics

Data on demographic characteristics (i.e., age, sex, height, body weight) and training characteristics (weeks of training, amount of training times per week and average distance per training) were collected using an online questionnaire which was sent to the participants 1–3 days prior to the road race. 

#### 2.4.2. Race Characteristics

Race finish times and pace of the participants were collected via the Seven Hills Run foundation that used the MyLaps Event timing system, Nijmegen, the Netherlands (https://www.mylaps.com/) (accessed on 4 December 2018). Furthermore, participants running with a heart rate monitor were asked to share their recordings with the research team. This approach allowed us to collect average heart rate and determine exercise intensity, defined as a percentage of the predicted maximal heart rate (208–0.7 *age) [31], in a subset of our study population.

#### 2.4.3. Subjective Muscle Soreness

The location and intensity of lower extremity muscle soreness was determined at baseline and 1 day, 2 days and 3 days post-race using the BPI-SF [30] which included a validated NPRS (a segmented numeric version of the visual analog scale) [32] where participants could mark a pain score between no pain at all (NPRS = 0) and extremely painful (NPRS = 10). NPRS scores of 1–5 were considered as mild pain, 6–7 as moderate pain and ≥8 as severe pain [33]. The questionnaires could be filled out between 18 to 34 h post-exercise (1 day post-race), 42 to 58 h post-exercise (2 days post-race) and 66 to 82 h post-exercise (3 days post-race). Participants were asked to score their least, maximal and average soreness of the last 24 h.

#### 2.4.4. Objective Muscle Soreness

In the subgroup of 149 participants, we measured sensitivity to muscle pain with a digital pressure algometer with a 1.0 cm^2^ probe (Wagner instruments, Force FX, Greenwhich, CT, USA). The reliability and validity of the algometer has been established previously [34,35,36]. Measurements were performed between 25 to 48 h post-race. The time of the measurement was recorded. Measurements were performed at the rectus femoris, vastus lateralis and vastus medialis oblique of the dominant leg. Two trained and experienced researchers performed these measurements and applied force on the approximate mid-point of the muscle via the probe until the participant indicated pain or discomfort (pressure pain threshold). At this point the force value (Newton) was recorded. All measurements were taken in triplicate from the dominant thigh in a seated position with the knee at a 90° angle after a familiarization session at the non-dominant leg.

#### 2.4.5. Measures that Could Influence Muscle Soreness

We asked participants to report any treatment (e.g., muscle cream or having a massage or a bath), medications or other nutritional supplementation for pain relief or other reasons that may have interacted with our primary outcome. Furthermore, subjects were queried about the performance of sports activities in the 3 days post-race. 

#### 2.4.6. Muscle Damage Biomarkers

In the subgroup of 149 participants, a non-fasted venous blood sample was drawn 24 to 48 h post-race from an antecubital vein into lithium heparin tubes, within half an hour centrifuged (1300× *g* at 20 °C for 10 min) and stored at −80 °C until analysis. Plasma CK and LDH were measured using the Siemens Atellica^TM^ Solution system (Siemens Healthcare Diagnostics Inc., Tarrytown, NY, USA). Analysis were performed by trained technicians using standard operating procedures, on a single day using the same calibration and set-up to minimize variation.

#### 2.4.7. Dietary Intake

Habitual dietary intake was assessed in the week prior to the race using a repeated 24 h recall, which is a validated method to assess the amount and distribution of habitual protein intake [37]. Participants were instructed to fill out their dietary intake of the previous day in the web-based program Compl-eat. Portion sizes were documented in household measures, whereby frequently used household measures were subsequently quantified with standard portion sizes. Two recall days were randomized over the week, with the restriction that no participant was assigned to two identical week days or two weekend days, to be able to determine the average of the two days that represented the habitual dietary intake. Moreover, participants were asked to report their dietary intake of the day of the race and 1 day and 2 days post-race in the same program (excluding the supplements). The 24 h recalls were then checked and coded by dieticians. The dietary intake was calculated using the Dutch Food Composition Database of 2016 [38].

Alcohol consumption may negatively alter protein synthesis and therefore may impair recovery of muscle damage [39]. Alcohol was derived in g per day of pure alcohol. For the day of the race, participants were classified as non-drinkers (<2 g/day), low drinkers (women: 2.06–20.86 g/day and men: 2.06–34.86 g/day), moderate drinkers (women: 20.87–48.86 g/day, men: 34.87–63.05 g/day) and high drinkers (women: ≥48.87 g and men: ≥63.06 g/day), based on the American guidelines of one alcoholic drink-equivalent of 14 g of pure alcohol per day [40]. 

### 2.5. Statistical Analysis

Based on a Type I- error of 0.05 and a power of 80% we calculated (G-power, version 3.1.2, University of Dusseldorf, Dusseldorf, Germany) that 176 participants per study arm were needed to find an expected 0.6 lower muscle soreness after protein supplementation compared to carbohydrate supplementation. To accommodate for non-compliance or drop-outs (15%) we needed to include 203 participants per study arm. 

Statistical analyses were performed using SPSS 22.0 software (IBM SPSS Statistics for Windows, Version 22.0 IBM Corp., Armonk, NY, USA). According to the CONSORT guidelines, primarily an intention-to-treat analysis (ITT) was performed and a per-protocol (PP) analysis was performed alongside to enable the influence of any missing data to be investigated [41]. The PP analysis included per day the participants that filled out the questionnaire and reported to be perfectly compliant (i.e., 250 mL post-exercise, 250 mL with breakfast and 500 mL prior to sleep) until the analyzed day (one, two or three days post-race). Both analyses were performed to show that the results of the actual effect of the study protocol (PP analyses), as well as to evaluate the feasibility of a certain protocol in a real-life setting and the results in that setting (ITT analysis). Muscle soreness of 1 day post-race was the primary outcome, and the 2 and 3 days post-race muscle soreness were secondary outcomes. All continuous variables were visually inspected and tested for normality with the Shapiro-Wilk test. Participant characteristics were displayed as mean ± SD or median (interquartile range (IQR)) for parametric and non-parametric continuous variables, respectively. Except for the non-parametric NPRS which was given as mean ± SD. Categorical variables were given as number of participants with percentages. Baseline characteristics and muscle soreness and other muscle damage markers were compared between groups by means of an independent-samples *t*-test or a Mann-Whitney U test for parametric or non-parametric continuous variables, respectively and with a chi-square or Fisher’s exact test (with cell-counts < 5) for categorical variables. A repeated measures ANOVA was performed to assess the interaction effect (time × treatment) on the subjectively measured muscle soreness for the PP analyses. Because no between-group differences were found for baseline characteristics, habitual dietary intake and undertaken measures that could influence muscle soreness, no variables were added as a confounder in the main analysis. The level of significance was set at *p* < 0.05 (two-sided).

## 3. Results

### 3.1. Participants

For this study, 419 participants were screened and 323 participants (aged 44 ± 11, 56% men) were included and randomly allocated to the protein or CHO placebo group. There were no differences between the protein and placebo group for any of the baseline characteristics, including the training status of the participants (Table 1). In total, 9 participants could not participate in the running event due to illness or injury or did not collect the supplements after the race, resulting in 157 participants in the protein group and 157 participants in the placebo group. After the race, 25 participants were lost to follow-up and 74 participants did not completely comply with the intervention. For each day the analyzed number of participants per group for the ITT and PP analysis are described in Figure 1. 

### 3.2. Race Characteristics

The 15 km run was finished by 319 participants with an average speed of 12.1 ± 2.2 km/h at an exercise intensity of 94 ± 6% of HRmax. No differences were observed between the protein and placebo group (Table 1).

### 3.3. Dietary Intake

Habitual protein intake was comparable between the protein and placebo group (1.11 ± 0.33 g/kg/d versus 1.17 ± 0.37 g/kg/d, *p* = 0.21) (Table 1), with 83% and 86% having a protein intake of ≥0.8 g/kg/d (*p* = 0.54) [42]. In contrast, only 31% of the protein group and 42% of the placebo group (*p* = 0.081) reached the protein recommendation for endurance athletes (≥1.2 g/kg/d) [43]. Daily energy, macronutrient and alcohol intake did not differ between groups at baseline (Table 1) and at 1 and 2 days post-race (Table 2). On the race day no between-group differences were found in the dietary intake before running and after running (Appendix A). After finishing the race, the number of non-drinker versus low, moderate and high drinkers were not significantly different between the protein and placebo group (*p* = 0.29, Appendix A).

Following supplementation, protein intake among compliant participants was 1.94 ± 0.43 g/kg/d protein on race day, 1.97 ± 0.44 g/kg/d at 1 day post-race and 1.89 ± 0.40 g/kg/d at 2 days post-race. Whereas the placebo group consumed 1.10 ± 0.40 g/kg/d protein on race day, 1.09 ± 0.39 g/kg/d at 1 day post-race and 1.12 ± 0.34 g/kg/d at 2 days post-race (Table 2).

### 3.4. Compliance

A total of 157 participants of the protein group and 157 participants of the placebo group received and consumed the supplement immediately after finishing. Participants were instructed to use 500 mL of the received supplement prior to sleep and 250 mL with breakfast and 137 (87%) participants of the protein group and 139 (89%) of the placebo group reported they used these amounts at day 1. Of these participants, 124 (91%) participants of the protein group and 127 (91%) participants of the placebo group reported compliance on day 2. Of these participants, 105 (85%) participants of the protein group and 109 (86%) participants of the placebo group reported compliance on day 3 (Figure 1).

### 3.5. Adverse Events

Four participants dropped out of which three participants had gastrointestinal complaints that were related to the supplement (67% protein group) and one participant became ill (0% protein group, Figure 1). Furthermore, another 13 participants reported adverse events but did not drop out; eight participants had supplement-related gastrointestinal complaints (50% protein group) and five participants reported illness (80% protein group). 

### 3.6. Muscle Soreness and Damage Markers

#### 3.6.1. Subjective Muscle Soreness

Baseline ratings of perceived muscle soreness (NPRS) were not different between the protein and placebo group (Table 3). Based on the ITT analysis, no significant differences between the protein and the placebo group were reported at one to three days post-race within NPRS scores (Figure 3). Furthermore, categorical analysis revealed no differences between groups at 1 day post-race (protein group: 25% no pain, 61% mild pain, 12% moderate pain and 3% severe pain versus placebo group: 31% no pain, 53% mild pain, 13% moderate pain and 3% severe pain (*p* = 0.53). On the contrary, based on the PP analysis, the protein group reported significantly higher ratings of perceived lower extremity muscle soreness compared to the placebo group (NPRS: 2.96 ± 2.27 versus 2.46 ± 2.38, *p* = 0.039, Figure 3) at 1 day post-race, but no differences across groups were found at 2- and 3-days post-race (Appendix A). We found no significant time*treatment interaction effect in a repeated measures ANOVA (*p* = 0.10). 

#### 3.6.2. Objective Muscle Soreness

Based on the ITT analysis, lower pressure pain thresholds were endured for the rectus femoris (*p* = 0.034) and the vastus medialis (*p* = 0.007) among participants of the protein group compared to the placebo group (Table 4), representing higher quadriceps soreness within the protein group (Figure 4). The timing of the objective muscle soreness measurement was not significantly different between groups (*p* = 0.55). The PP analysis revealed similar findings as the ITT analysis (Appendix A).

#### 3.6.3. Blood Analyses

Concentrations of the muscle damage markers CK (protein: 265 (171–495) unit/liter (U/L) versus placebo: 366 (196–508) U/L, *p* = 0.26) and LDH (protein: 181 (144–200) U/L versus placebo: 170 (147–196) U/L, *p* = 0.60) did not differ between the protein group compared to the placebo group based on the ITT analysis (Figure 5) at 1- and 2-days post-race. The timing of post-race blood sampling was not significantly different across groups (all *p* > 0.05). The PP analysis revealed similar findings as the ITT analysis (Appendix A).

### 3.7. Measures That Could Influence Muscle Soreness

No differences were found between groups for muscle soreness countermeasures. One day post-race, four participants (3%) of the protein group versus six participants (5%) of the placebo group performed actions to diminish the muscle soreness (*p* = 0.35). Two and three days post-race, three (3%) and 1 (1%) participants of the protein group versus 1 (1%) and zero participants (0%) of the placebo group performed actions to diminish the muscle soreness (*p* = 0.40 and *p* = 0.54, respectively). One day post-race, 36 participants (27%) of the protein group and 34 participants (25%) of the placebo group performed sports activities (*p* = 0.70). On 2 days post-race, 45 participants (39%) of the protein group and 53 participants (44%) of the placebo group performed sports activities (*p* = 0.40).

## 4. Discussion

In this double-blind randomized placebo-controlled trial, we found no greater benefit of protein supplementation on muscle soreness and other muscle damage markers after a 15-km road race compared to the placebo group that received carbohydrate supplementation. Based on the ITT analysis, the protein group reported similar muscle soreness compared to the placebo group and similar muscle damage markers were observed. Within the placebo group however, higher quadriceps pain thresholds were found compared to the protein group. Moreover, with the per-protocol analysis the placebo group reported lower muscle soreness compared to the protein group, on one day post-race, but not on day 2 and 3 post-race. These findings indicate that three doses of post-exercise protein supplementation resulting in average protein intake of 1.94 ± 0.43 g/kg/d on race day, 1.97 ± 0.44 g/kg/d at one day post-race and 1.89 ± 0.40 g/kg/d at 2 days post-race are not more preferable than carbohydrate supplementation in the placebo group that consumed 1.10 ± 0.40 g/kg/d protein on race day, 1.09 ± 0.39 g/kg/d at one day post-race and 1.12 ± 0.34 g/kg/d at two days post-race to reduce exercise-induced muscle soreness and other damage markers in recreational endurance athletes that already have a sufficient protein intake.

To reduce muscle damage a positive muscle protein balance is required to induce growth and repair of contractile protein [4]. It has been suggested that sufficient daily caloric and protein intake is most crucial for a positive muscle protein balance [44]. Perhaps the habitual protein intake (protein group: 1.11 ± 0.33 g protein per kg/d, placebo group: 1.17 ± 0.37 g/kg/d) was already sufficient among our study participants and thus the protein supplementation could not result in a superior effect above their normal diet. A protein intake of 0.8 g/kg/d is recommended for the general population [42], whereas endurance athletes should consume 1.2–1.4 g/kg/d [43]. We found a tendency (*p* = 0.081) towards a larger number of participants in the placebo group (42%) with a habitual protein intake ≥ 1.2 g/kg/d compared to the number of participants in the protein group (31%). Carbohydrates can also improve the muscle protein balance because of the increased insulin secretion which inhibits protein breakdown [27,28]. Moreover, co-ingestion of carbohydrates with protein synergistically has been shown to augment insulin secretion and improves muscle protein balance because of the increased amino acid availability and delivery to the muscle compared with ingestion of carbohydrates alone [27,28,45]. Indeed, after a 90-min cycling bout the most beneficial effect on the muscle damage marker CK were present in the protein group, but CHO supplementation also significantly attenuated CK compared to the group that received no calories [7]. The independent and synergistic effects of the carbohydrate supplementation with the habitual protein intake on muscle protein balance could have contributed to the reduced muscle soreness in our placebo group. 

Another explanation for the lack of beneficial effects of protein supplementation on muscle soreness and other damage markers could be that within our study, the protein supplementation was only given post-race and not pre-race. Out of four studies that found a positive effect of protein supplementation on muscle damage markers and soreness [6,7,8,9], three studies had a protein supplementation protocol in which they started the supplementation before the onset (varying from three weeks to 5 min) of the endurance exercise [6,7,8]. The other study with positive findings had a regimen in which every 15 min during the endurance exercise the participants received supplementation [9]. Within the studies in which protein supplementation did not attenuate muscle soreness and other damage markers [10,11,12], protein was only given after the endurance exercise [11,12] or during exercise and after finishing [10]. Thus, protein supplementation before the endurance exercise might be more beneficial to attenuate exercise-induced muscle soreness.

Finally, the relatively low levels of exercise-induced muscle soreness and other muscle damage markers could contribute to the absent superior effects of protein supplementation. Peak levels of perceived muscle soreness were found one day post-race and were on average 2.96 ± 2.27 in the protein group and 2.46 ± 2.38 in the placebo group. These low muscle soreness values could be caused by the repeated bout effect phenomenon, that decreases muscle damage from subsequent eccentric bouts after one single bout [13], since our study participants train on average two times a week for 48 weeks a year Although other studies found higher peak scores with values varying between 4.9 ± 0.5 and 5.9 ± 0.4 at 72 h post-exercise [11,12], our participants represented a cohort of trained amateur athletes who were exposed to a high exercise intensity (94 ± 6% of maximum predicted heart rate) but only for a moderate duration (76 ± 13 min). Indeed, a lower muscle soreness was reported in the protein versus the placebo group (protein: 4.76 versus placebo (containing CHO): 6.50) at 24 h after performing repeated-sprint exercise. Alternatively, endurance exercise of longer duration and/or with more eccentric load on the muscle, might have induced greater muscle soreness and damage. A study performed in marathon participants found CK levels of 545 U/L in the placebo group and 233 U/L in the whey-protein group after one day (*p* < 0.016) [8], whereas we found lower CK levels in the placebo group (346 U/L) and in the protein group (285 U/L). These observations are in agreement with an earlier study in which mild muscle damage (CK 4 h post-exercise: 162 U/L (protein) and 125 U/L (placebo)) was present after a cycling bout of 1.5 h at 70% of VO_2_max, which was also not affected by protein supplementation compared to the carbohydrate drink [46,47]. Maybe protein only attenuate substantial exercise-induced muscle damage and soreness, whereas carbohydrates are sufficient to attenuate mild exercise-induced muscle damage and soreness. Indeed, stimulating faster glycogen repletion through carbohydrate supplementation may positively affect selected markers of muscle damage, such as muscle soreness [48,49] and muscle contractile function [50]. Future studies are warranted to assess if and how carbohydrate supplementation can attenuate the degree of muscle damage. 

## 5. Practical Relevance

After a 15-km running race, mostly only mild muscle soreness (NPRS: 1–5) was induced and only 17% of the protein group and 15% of the placebo group reported moderate (NPRS: 6–7) to severe (NPRS: ≥ 8) pain at 1 day post-race. Our findings indicate that in adults that have a sufficient protein intake, post-race carbohydrate-supplementation is more effective in reducing mild exercise-induced muscle soreness in the first 24 h compared to post-race protein supplementation. 

## 6. Limitations

Although our targeted sample size was not reached (resulting in reduced power of 71% on day 1, 66% on day 2 and 60% on day 3 to find a 0.6 lower pressure pain threshold score), we found statistically significant differences between groups. The sample size within our study was considerably greater compared to previous studies [6,7,8,9,10,11,12]. We did not measure muscle damage markers in the blood and objective muscle soreness before the race. Previous studies have demonstrated, however, that pre-exercise CK levels are typically low (100 U/L) and similar between protein and placebo groups [7,12]. We assume that within our large randomized group of participants no differences in muscle damage markers in the blood and objective muscle soreness were present at baseline, especially since the NPRS scores exhibited also no differences between groups at baseline. Another limitation of our study was that objective measurements of muscle soreness were only performed in a subgroup of 149 participants. However, outcomes from our subgroup measurements were in alignment with the NPRS scores that were collected in the whole study population. Finally, based on this study, we cannot conclude whether post-exercise protein or carbohydrate supplementation is better compared to no supplementation at all or whether protein supplementation positively influences muscle function.

## 7. Conclusions

While with the ITT analysis no differences were found between groups, in the per-protocol analysis one day post-race lower muscle soreness and higher pain threshold was found following carbohydrate supplementation versus protein supplementation among recreational athletes completing a 15-km road race. These findings suggest that protein supplementation directly post-exercise and with breakfast and prior to sleep until 3 days post-race is not superior, and potentially inferior, to reduce endurance exercise-induced muscle soreness and other damage markers in the first 24 h following a muscle damaging activity, compared to carbohydrate supplementation in recreational endurance athletes that already have an adequate habitual protein intake and report mild to moderate post-exercise muscle soreness and other damage markers.

## Figures and Tables

**Figure 1 nutrients-13-00858-f001:**
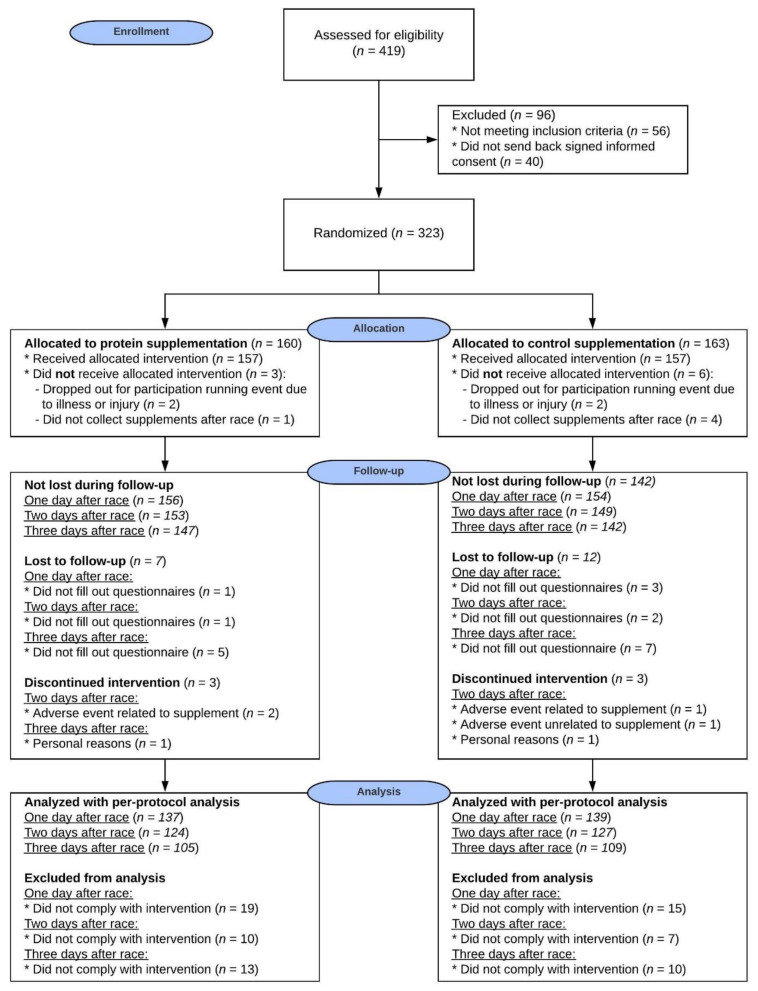
CONSORT Flow diagram illustrating the movement of the participants through the study.

**Figure 2 nutrients-13-00858-f002:**
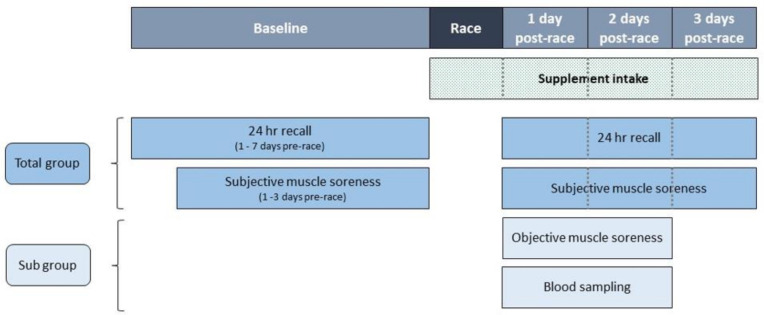
Schematic overview of the study procedure.

**Figure 3 nutrients-13-00858-f003:**
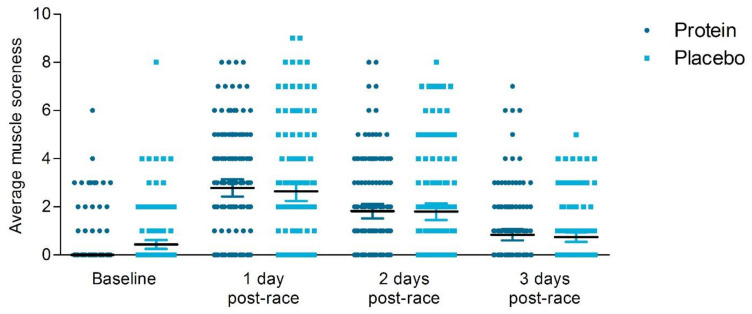
Ratings of perceived lower extremity muscle soreness of the past 24 h (Numeric Pain Rating Scale). The ratings are shown from baseline, 1 day post-race, 2 days post-race and 3 days post-race for the protein group (● dark blue circles) and the placebo group (▪ light blue squares) with the median and interquartile range (IQR), based on intention-to-treat analysis. The protein group reported similar ratings of perceived lower extremity muscle soreness at 1–3 day post-race compared to the placebo group (*p* > 0.05).

**Figure 4 nutrients-13-00858-f004:**
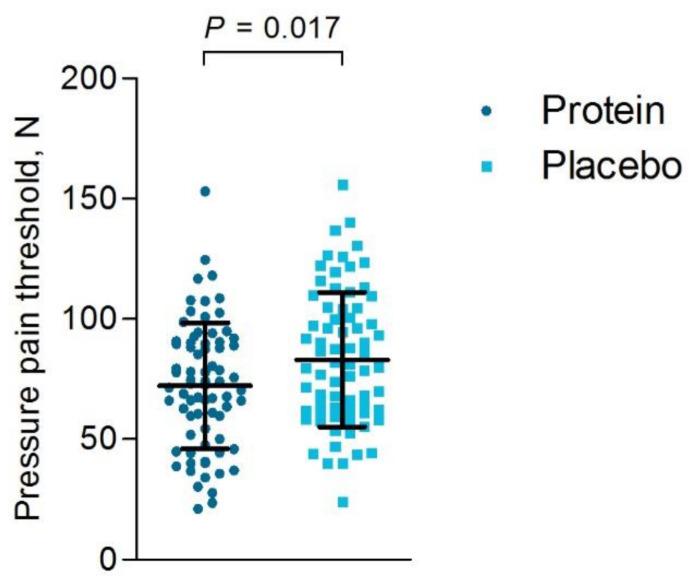
Pressure pain thresholds one and two days post-race in N (newton). The pressure pain thresholds are shown for the protein group (● dark blue circles) and the placebo group (▪ light blue squares) with the mean and standard deviation (SD), based on intention-to-treat analysis. Within the protein group significantly lower pressure thresholds on the quadriceps could be endured compared to the placebo group, representing higher quadriceps soreness compared to the placebo group (*p* = 0.017).

**Figure 5 nutrients-13-00858-f005:**
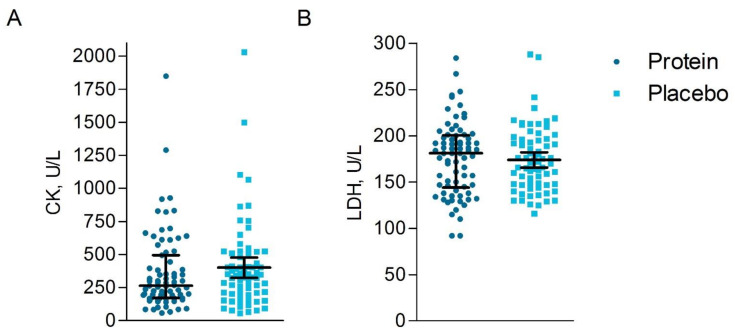
Muscle damage markers measured one and two days post-race. The muscle damage markers CK (creatine kinase) (**A**) and LDH (lactate dehydrogenase) (**B**) in U/L (units/liter) are shown for the protein group (*n* = 75) (● dark blue circles) and the placebo group (*n* = 74) (▪ light blue squares) with the median and interquartile range (IQR), based on intention-to-treat analysis. Similar concentrations of the muscle damage markers CK and LDH were found between groups.

**Table 1 nutrients-13-00858-t001:** Baseline characteristics, based on ITT.

	Total Group *n* = 323	Protein *n* = 160	Placebo *n* = 163	*p*-Value
Demographics				
Age, years	44 ± 11	44 ± 12	44 ± 10	0.64
Men, *n* (%)	181 (56)	88 (55)	93 (57)	0.71 ^§^
Weight, kg	71.6 ± 11.0 ^a^	71.4 ± 10.5 ^b^	71.4 ± 10.5 ^c^	0.89
BMI, kg/m^2^	22.7 ± 2.2 ^a^	22.6 ± 2.2 ^b^	22.8 ± 2.2 ^c^	0.51
Running training				
Weeks of training, nr	48 (40–52) ^a^	50 (40–52) ^b^	48 (40–52) ^c^	0.65 ^‡^
Number of times per week	2 (2–3) ^a^	2 (2–3) ^b^	2 (3–3) ^c^	0.76 ^‡^
Average distance per training, km	10 (8–12) ^a^	10 (8–12) ^b^	10 (8–12) ^c^	0.61 ^‡^
Race				
Finish time, min	76 ± 13 ^d^	77 ± 13 ^e^	76 ± 13 ^f^	0.88
Running speed, km/hr	12.1 ± 2.2 ^d^	12.2 ± 2.2 ^e^	12.0 ± 2.1 ^f^	0.48
Average heart rate, bpm	168 ± 12 ^g^	170 ± 10 ^h^	166 ± 13 ^i^	0.13
Exercise intensity, %	94 ± 6 ^g^	95 ± 5 ^h^	94 ± 7 ^i^	0.48
Diet				
Energy intake, kcal	2031 ± 632 ^j^	2017 ± 622 ^k^	2046 ± 644 ^l^	0.69
Protein intake, g	80.9 ± 26.6 ^j^	79.9 ± 26.5 ^k^	82.0 ± 26.8 ^l^	0.49
Protein intake, g/kg/d	1.14 ± 0.35 ^m^	1.11 ± 0.33 ^n^	1.17 ± 0.37 ^o^	0.21
Animal protein, %	52.0 ± 14.2 ^j^	52.0 ± 14.1 ^k^	51.9 ± 14.3 ^l^	0.94
Plant protein, %	47.7 ± 14.2 ^j^	47.9 ± 14.0 ^k^	47.5 ± 14.5 ^l^	0.82
Protein, en%	16.7 ± 4.1 ^j^	16.6 ± 4.0 ^k^	16.8 ± 4.1 ^l^	0.70
Fat intake, en%	33.3 ± 8.0 ^j^	33.2 ± 8.7 ^k^	33.4 ± 7.3 ^l^	0.79
Carbohydrate intake, en%	45.7 ± 9.0 ^j^	46.1 ± 10.4 ^k^	45.3 ± 7.4 ^l^	0.41
Alcohol intake, g	0.0 (0.0–9.6)	0.0 (0.0–9.6)	0.0 (0.0–9.3)	0.43 ^‡^

Data are presented as mean ± SD or median (interquartile range (IQR)). BMI, body mass index; bpm, beats per minute; en%, energy percentage; ITT, Intention-to-treat analysis. ^a^
*n* = 244, ^b^
*n* = 116, ^c^
*n* = 128, ^d^
*n* = 319, ^e^
*n* = 158, ^f^
*n* = 161, ^g^
*n* = 78, ^h^
*n* = 36, ^I^
*n* = 42, ^j^
*n* = 294, ^k^
*n* = 150, ^l^
*n* = 144, ^m^
*n* = 229, ^n^
*n* = 110, ^o^ n = 119. *p*-values for differences between groups were derived by independent samples t-test unless indicated otherwise. § Derived by chi-square test. ‡ Derived by Mann-Whitney U-test.

**Table 2 nutrients-13-00858-t002:** Habitual dietary intake of participants in the protein and placebo group (disregarding supplements), based on ITT.

	Day of the Race			1 Day Post-Race				2 Days Post-Race	
	Protein *n* = 134	Placebo *n* = 127	*p*-Value	Protein *n* = 119	Placebo *n* = 115	*p*-Value	Protein *n* = 119	Placebo *n* = 112	*p*-Value
Energy intake, kcal	2293 ± 790	2277 ± 697	0.86	1916 ± 596	1942 ± 717	0.76	1807 ± 580	1910 ± 624	0.20
Protein intake, g	76.5 ± 30.8	79.4 ± 36.1	0.49	75.4 ± 26.0	75.4 ± 29.0	0.99	73.6 ± 27.6	77.5 ± 26.7	0.27
Protein intake, g/kg/d	1.05 ± 0.38 ^a^	1.10 ± 0.40 ^b^	0.43	1.07 ± 0.38 ^c^	1.09 ± 0.39 ^d^	0.78	1.05 ± 0.36 ^e^	1.12 ± 0.34 ^f^	0.17
Animal protein, %	49.4 ± 18.0	50.0 ± 17.3	0.77	49.7 ± 17.8	51.4 ± 18.1	0.47	51.0 ± 18.2	53.3 ± 15.3	0.30
Plant protein, %	50.6 ± 18.1	49.9 ± 17.3	0.77	50.3 ± 17.8	48.4 ± 17.6	0.41	49.0 ± 18.2	46.7 ± 15.4	0.31
Protein, en%	13.9 ± 4.4	14.2 ± 4.2	0.62	16.3 ± 4.0	16.4 ± 5.1	0.77	16.8 ± 4.1	17.0 ± 4.1	0.73
Fat intake, en%	30.5 ± 9.2	31.3 ± 8.7	0.49	33.3 ± 9.4	34.9 ± 10.5	0.22	32.5 ± 9.3	34.6 ± 8.5	0.07
Carbohydrate intake, en%	48.7 ± 10.5	48.1 ± 10.0	0.65	46.1 ± 9.7	45.1 ± 9.4	0.44	46.2 ± 10.1	44.6 ± 8.7	0.22
Alcohol intake, g	7.9 (0.0–25.9)	0.0 (0.0–18.9)	0.25 ^‡^	0.0 (0.0–0.0)	0.0 (0.0–0.3)	0.48 ^‡^	0.0 (0.0–0.0)	0.0 (0.0–0.0)	0.58 ^‡^

Data are presented as mean ± SD or median (interquartile range (IQR)). en%, energy percentage; ITT, Intention-to-treat analysis. ^a^
*n* = 97, ^b^
*n* = 106, ^c^
*n* = 89, ^d^
*n* = 99, ^e^
*n* = 95, ^f^
*n* = 95. *p*-values for differences between groups were derived by independent samples t-test unless indicated otherwise. ‡ Derived by Mann-Whitney U-test.

**Table 3 nutrients-13-00858-t003:** Baseline and post-race NPRS scores for perceived lower extremity muscle soreness, based on ITT.

		Protein	Placebo	*p*-Value
		*n* = 150	*n* = 150	
Baseline	Worst muscle soreness	0.67 ± 1.61	0.66 ± 1.58	0.93
	Least muscle soreness	0.27 ± 0.78	0.20 ± 0.69	0.53
	Average muscle soreness past 24 h	0.45 ± 1.08	0.44 ± 1.14	0.96
		*n* = 155	*n* = 152	
1 day post-race	Worst muscle soreness	3.83 ± 2.84	3.67 ± 3.01	0.61
	Least muscle soreness	1.65 ± 1.93	1.43 ± 1.96	0.16
	Average muscle soreness past 24 h	2.79 ± 2.26	2.53 ± 2.47	0.17
		*n = 129*	*n* = 125	
2 days post-race	Worst muscle soreness	2.54 ± 2.38	2.54 ± 2.71	0.69
	Least muscle soreness	1.19 ± 1.58	1.17 ± 1.68	0.49
	Average muscle soreness past 24 h	1.82 ± 1.88	1.8 ± 2.16	0.44
		*n* = 148	*n* = 144	
3 days post-race	Worst muscle soreness	1.26 ± 1.96	1.08 ± 1.70	0.74
	Least muscle soreness in	0.49 ± 1.01	0.42 ± 0.80	0.78
	Average muscle soreness past 24 h	0.84 ± 1.42	0.74 ± 1.20	0.76

Data are presented as mean ± SD. ITT, Intention-to-treat analysis. NPRS; numeric pain rating scale. *p*-values for differences between groups were derived by Mann-Whitney U-test.

**Table 4 nutrients-13-00858-t004:** Pressure pain thresholds with a strain gauge algometer measured 1–2 days post-race, based on ITT.

	Protein *n* = 74 ^a^	Placebo *n* = 74	*p*-value
M. vastus lateralis, N	73.8 ± 29.4	82.7 ± 33.5	0.088
M. rectus femoris, N	81.5 ± 31.0	92.4 ± 31.1	0.034
M. vastus medialis, N	61.6 ± 25.4	74.0 ± 30.0	0.007
Average 3 muscles, N	72.3 ± 26.1	83.1 ± 28.0	0.017

Data are presented as mean ± SD. M, musculus; ITT, Intention-to-treat analysis. ^a^ One participant was excluded due to an invalid measurement. *p*-values for differences between groups were derived by independent samples *t*-test.

## Data Availability

The datasets generated and/or analyzed during the current study are not publicly available due to the fact that on the base of age, sex and finish time identities of participants can be tracked down, but are available from the corresponding author on reasonable request.

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
