# Peer review of "The Effect of Protein Supplementation versus Carbohydrate Supplementation on Muscle Damage Markers and Soreness Following a 15-km Road Race: A Double-Blind Randomized Controlled Trial"

_nutrients, 2021, doi:10.3390/nu13030858_

Round 1

Reviewer 1 Report

This research was to compare a protein supplementation with CHO on attenuating intensive endurance race induced muscle soreness and muscle damage post exercise. The research is interesting and the manuscript is in good shape, but there are some major concerns that the authors should consider and address.

  1. In the second paragraph of the introduction, it was unclear what population the previous research has focused on. Did they target the recreational athletes, younger adult, elderly, or others?
  2. In the introduction, please expand more to summarize optimal dose of protein supplementation in different population, especially in the recreational athletes. In addition, authors only introduced the optimal dose for muscle protein synthesis, but it is unclear to the impact of muscle protein breakdown.
  3. CHO increases insulin secretion. Insulin has been known to inhibit protein breakdown. If you use CHO as a placebo, the impact of CHO should also be described in the introduction. In addition, so many research have found that protein was primarily used to stimulate protein synthesis by activating mTOR signaling pathway, but it (especially whey protein) did little impact on the muscle protein breakdown inhibition. Therefore, authors should be more clear to what gap this research tried to fill in. It is needed both in the introduction and discussion.
  4. Lines 58-60: Because the milk you used contained 80% casein and 20% whey protein, it is necessary to state/introduce how exactly the two types of proteins modulate the protein accretion differently. For example, casein is primarily attenuate protein breakdown whereas whey protein stimulate protein synthesis, etc.
  5. Section 2.3: It is important to provide rationale why you used milk containing 80% casein and 20% whey protein. Also, what is the reason 40g protein should be consumed prior to sleep if the optimal dose most recreational athletes have taken daily? Typically, it is recommended that casein was supplied before sleep and whey protein was supplied immediately after the exercise. Therefore, please provide rationale either in this section or in the introduction on the dose, timing, and percentages. You mentioned it a little bit in the second paragraph of the introduction, but it is not sufficient to convince audience.
  6. What is the reason that you only measure objective soreness and took blood on sub-group?
  7. It is unclear why both ITT and PP analyses were used to deal with the missing data. Which result is more reliable when the results on subjective muscle soreness at day 1 were different between two analyses?
  8. Because of the relatively large sample size in this study, the normality is robust and all analyses could be done with the parametric analyses.
  9. It appears that authors only examined the difference between the groups at each time point, but did not look at if the subjective muscle soreness changed over the four time points. If this is the case, please run repeated measures ANOVA to look at if the muscle soreness changed over time. I am just curious whether or not the protein group increased the protein synthesis in order to speed up muscle repair from day 1 through day 3.

Author Response

February 24th 2021

We are pleased to submit the revised version of our research article entitled “The effect of protein supplementation versus carbohydrate supplementation on muscle damage markers and soreness following a 15-km road race: a double-blind randomized controlled trial” (nutrients-1095772).

We thank the reviewers for their thorough examination of the manuscript and revision and for their valuable comments and suggestions.

A point-by-point response to the comments from the reviewers, including revisions made to the manuscript in red, can be found below. Additionally, the revisions made to the manuscript are shown by track changes in the manuscript.

Revision note

Manuscript ID: nutrients-1095772

Manuscript title: The effect of protein supplementation versus carbohydrate supplementation on muscle damage markers and soreness following a 15-km road race: a double-blind randomized controlled trial

Reviewer 1

This research was to compare a protein supplementation with CHO on attenuating intensive endurance race induced muscle soreness and muscle damage post exercise. The research is interesting and the manuscript is in good shape, but there are some major concerns that the authors should consider and address.

  1. In the second paragraph of the introduction, it was unclear what population the previous research has focused on. Did they target the recreational athletes, younger adult, elderly, or others?

Thank you for pointing this out. The described studies were performed in males and females aged around 22 years with varying fitness levels, ranging from recreational to elite level. We added this information in line 51-58.

‘’Small-scale studies including few participants (n=9 to n=16, age ~22 years) reported positive effects of protein supplementation following high-intensity endurance exercise, on muscle damage markers and muscle soreness on subsequent days in healthy men with fitness levels ranging from recreational to elite level (6-9). However, other studies failed to replicate these findings in men and women with varying fitness levels and reported no difference in muscle damage markers and muscle soreness between the protein supplementation group and placebo group (10-12).’’

  1. In the introduction, please expand more to summarize optimal dose of protein supplementation in different population, especially in the recreational athletes. In addition, authors only introduced the optimal dose for muscle protein synthesis, but it is unclear to the impact of muscle protein breakdown.

We have added the required information to the second paragraph of the introduction. Protein ingestion stimulates muscle protein synthesis (MPS) and inhibits muscle protein breakdown (MPB) rates, resulting in net muscle protein accretion during the acute stages of post-exercise recovery [52]. In addition, a single session of exercise was shown to stimulate MPS rates and, to a lesser extent, muscle protein breakdown rates [5, 51]. Therefore, it is common to mainly focus on muscle protein synthesis, line 62-74.

'’A systematic review concluded that more attention should be given to the effects of different protein sources and timing of supplementation on muscle damage, soreness and recovery of muscle function and physical performance (4). Both casein and whey protein in milk protein contain all essential amino acids required to effectively stimulate muscle protein synthesis (46, 50). Moreover, to support muscle protein synthesis an effective pattern of daily protein intake distribution is to provide at least 20 g of protein with each main meal with no more than 4–5 h between each meal (13, 14, 47, 48) or after one-legged resistance type exercise (15). An extra opportunity during the day for protein intake is to take at least 40 g of protein prior to sleep to stimulate muscle protein synthesis until the next morning (16, 44-45, 49). These thresholds are often not reached by adults (17, 18).’’

  1. CHO increases insulin secretion. Insulin has been known to inhibit protein breakdown. If you use CHO as a placebo, the impact of CHO should also be described in the introduction. In addition, so many research have found that protein was primarily used to stimulate protein synthesis by activating mTOR signaling pathway, but it (especially whey protein) did little impact on the muscle protein breakdown inhibition. Therefore, authors should be more clear to what gap this research tried to fill in. It is needed both in the introduction and discussion.

In line 75-80 we introduce the possible beneficial effects of carbohydrates for reaching a positive muscle protein balance. Consumption of carbohydrates may, therefore, result in an improved muscle protein balance independent from protein intake. We explain that it is currently unknown how these proposed positive effects of protein and carbohydrates on the muscle protein balance translate in reducing exercise-induced muscle soreness and muscle damage.

‘’On the other hand, it has been shown that carbohydrates inhibit protein breakdown as well due to an augmented insulin secretion (36, 37). It is unknown how these proposed effects of protein and carbohydrates intake on muscle protein balance translate in reducing exercise induced muscle soreness and muscle damage and whether only increasing the protein intake after exercise is superior in reducing exercise-induced muscle soreness and muscle damage compared to carbohydrates only.’’

And added a sentence to our first explanation of our findings in our discussion, see line 414-415.

‘’Carbohydrates can also improve the muscle protein balance because of the increased insulin secretion which inhibits protein breakdown (36, 37). Moreover, co-ingestion of carbohydrates with protein synergistically has been shown to augment insulin secretion and improves muscle protein balance because of the increased amino acid availability and delivery to the muscle compared with ingestion of carbohydrates alone (35-37). Indeed, after a 90-min cycling bout the most beneficial effect on the muscle damage marker CK were present in the protein group, but CHO supplementation also significantly attenuated CK compared to the group that received no calories (7). The independent and synergistic effects of the carbohydrate supplementation with the habitual protein intake on muscle protein balance could have contributed to the reduced muscle soreness in our placebo group.‘’

  1. Lines 58-60: Because the milk you used contained 80% casein and 20% whey protein, it is necessary to state/introduce how exactly the two types of proteins modulate the protein accretion differently. For example, casein is primarily attenuate protein breakdown whereas whey protein stimulate protein synthesis, etc.

Whey protein has indeed faster digestion and absorption kinetics than casein, which results in a greater increase in postprandial plasma amino acid availability, thereby further stimulating muscle protein synthesis. Gorissen et al 2020 showed that the ingestion of milk protein resulted in intermediate peak exogenous phenylalanine appearance rates compared to whey protein and casein, with the time-to-peak being similar to casein ingestion (53). Whey and casein also markedly differ in their amino acid composition (13, 14, 16), but both casein and whey in milk protein contain all the (essential) amino acids required to effectively stimulate muscle protein synthesis as described in line 66-67.

‘’Both casein and whey protein in milk protein contain all essential amino acids required to effectively stimulate muscle protein synthesis (46, 50).’’

  1. Section 2.3: It is important to provide rationale why you used milk containing 80% casein and 20% whey protein. Also, what is the reason 40g protein should be consumed prior to sleep if the optimal dose most recreational athletes have taken daily? Typically, it is recommended that casein was supplied before sleep and whey protein was supplied immediately after the exercise. Therefore, please provide rationale either in this section or in the introduction on the dose, timing, and percentages. You mentioned it a little bit in the second paragraph of the introduction, but it is not sufficient to convince audience.

We have chosen for milk proteins, because both whey protein and casein have good anabolic properties due to their amino acid composition (50), and it has been shown for both casein and whey separately as well as for milk with 80% casein and 20% whey protein that the intake can stimulate post-exercise muscle protein synthesis rates (46).   
Besides the amount and type of ingested protein, the timing and distribution of protein ingestion throughout the day can modulate post-exercise muscle protein synthesis rates. An even distribution of total protein intake over the three main meals stimulates 24 h muscle protein synthesis rates more effectively than an unbalanced distribution in which the majority (>60%) of total daily protein intake is consumed at the evening meal (47). During 12 h of post-exercise recovery, an intermediate pattern of protein ingestion (4 x 20 g of protein every 3 hours) seems to increase muscle protein synthesis rates to a greater extent than the same amount of protein provided in less frequent but larger amounts (2 x 40 g every 6 hours), or in more frequent, smaller amounts (8 x 10 g every 6 hours) (48). Therefore, an effective pattern of daily protein intake distribution to support muscle protein synthesis is to provide at least 20 g of protein with each main meal with no more than 4–5 h between each meal.

An extra opportunity during the day for protein intake is prior to sleep.  It was investigated intensively that at least 40 g is needed prior to sleep in order to have a higher MPS in the morning (16, 44-45, 49). Moreover, Res (2012) also supplemented with whey protein and also showed stimulated muscle protein synthesis with whey protein and improved whole-body protein balance during postexercise overnight recovery (44). We explained the above in the article in line 66-74 of the introduction.

’Both casein and whey protein in milk protein contain all essential amino acids required to effectively stimulate muscle protein synthesis (46, 50). Moreover, to support muscle protein synthesis an effective pattern of daily protein intake distribution is to provide at least 20 g of protein with each main meal with no more than 4–5 h between each meal (13, 14, 47, 48) or after one-legged resistance type exercise (15). An extra opportunity during the day for protein intake is to take at least 40 g of protein prior to sleep to stimulate muscle protein synthesis until the next morning (16, 44-45, 49). These thresholds are often not reached by adults (17, 18).’’

  1. What is the reason that you only measure objective soreness and took blood on sub-group?

Our main goal of the study was to assess subjective measures of muscle soreness in a large group of participants, since most studies are only performed in small sample sizes. Furthermore, we wanted to confirm whether these outcomes correlated with objective measures of muscle soreness and muscle damage and invited study participants for this purpose for an additional study visit to our laboratory. Due to constraints in personnel and laboratory equipment (i.e. maximal number of blood drawings/day), this was only possible for a sub-group of participants. Nevertheless, this was only applicable to our secondary outcome measures and did not affect the primary research question.

  1. It is unclear why both ITT and PP analyses were used to deal with the missing data. Which result is more reliable when the results on subjective muscle soreness at day 1 were different between two analyses?

The benefit of performing a study in a real-life setting is that you can test whether study protocols are feasible to perform in real-life and whether certain protocols that have been studied in a lab-setting are also beneficial in a real-life setting. By doing both the ITT and PP analyses we were able to show that while you might find differences between groups when everybody exactly follows the protocol, it is often not feasible for participants to follow the protocol that strictly. When participants do not abide strictly to the protocol you might see other results as shown in the different result within the subjective muscle soreness at 1 day post-exercise for the ITT and PP analyses. Thus, when one wants to assess the effect of the study protocol the PP analyses is more reliable, but when translating it to a real-life setting so that more people can use it, the ITT gives a more reliable image of the efficacy of a certain protocol. Therefore, we felt that both analyses were interesting to show in the manuscript, as indicated in the manuscript in line 239-263.

‘’According to the CONSORT guidelines, primarily an intention-to-treat analysis (ITT) was performed and a per-protocol (PP) analysis was performed alongside to enable the influence of any missing data to be investigated (31). The PP analysis included per day the participants that filled out the questionnaire and reported to be perfectly compliant (i.e. 250 ml post-exercise, 250 ml with breakfast and 500 ml prior to sleep) until the analyzed day (one, two or three days post-race). Both analyses were performed to show that the results of the actual effect of the study protocol (PP analyses), as well as to evaluate the feasibility of a certain protocol in a real-life setting and the results in that setting (ITT analysis).’’

  1. Because of the relatively large sample size in this study, the normality is robust and all analyses could be done with the parametric analyses.

We indeed only used a parametric test for the pressure pain thresholds outcome and we used non-parametric tests for the perceived lower extremity muscle soreness and the muscle damage blood markers. We did this because the Kolmogorov-Smirnov and Shapiro-Wilk normality tests were both significant with P < 0.001 and the histograms also showed that no normal distributed data was obtained for these outcome parameters. Although we agree with the reviewer that with a large sample size normality is robust, we felt it was then still remarkable that for some outcomes we do see normally distributed data and for others not. Therefore, we choose to follow the outcomes of our normality testing and use both parametric and non-parametric testing where applicable.

  1. It appears that authors only examined the difference between the groups at each time point, but did not look at if the subjective muscle soreness changed over the four time points. If this is the case, please run repeated measures ANOVA to look at if the muscle soreness changed over time. I am just curious whether or not the protein group increased the protein synthesis in order to speed up muscle repair from day 1 through day 3.

We indeed looked at the outcomes on the different time points because we were interested in what happened at 1 day (primarily), 2 days and 3 days post-exercise (secondarily) individually. In the participants that filled in their average muscle soreness at baseline, 1 day post-race and 2 and 3 days post-race we performed a repeated measures ANOVA and we found no significant time*treatment effect (P = 0.10). We added this finding to result section in line 334-335 and added this to the statistical analyses in the methods section, line 259-260.

Methods:

‘’A repeated measures ANOVA was performed on the subjectively measured muscle soreness for the PP analyses.’’

Results

‘’We found no significant time*treatment effect in a repeated measures ANOVA (p = 0.10).’’

Mentioned references in response to Reviewer 1

  1. Levenhagen DK, Carr C, Carlson MG, Maron DJ, Borel MJ, Flakoll PJ. Postexercise protein intake enhances whole-body and leg protein accretion in humans. Med Sci Sports Exerc. 2002;34(5):828-37.
  2. Pasiakos SM, Lieberman HR, McLellan TM. Effects of protein supplements on muscle damage, soreness and recovery of muscle function and physical performance: a systematic review. Sports Med. 2014;44(5):655-70.
  3. Biolo G, Maggi SP, Williams BD, Tipton KD, Wolfe RR. Increased rates of muscle protein turnover and amino acid transport after resistance exercise in humans. Am J Physiol. 1995;268(3 Pt 1):E514-20.
  4. Coombes JS, McNaughton LR. Effects of branched-chain amino acid supplementation on serum creatine kinase and lactate dehydrogenase after prolonged exercise. J Sports Med Phys Fitness. 2000;40(3):240-6.
  5. Greer BK, Woodard JL, White JP, Arguello EM, Haymes EM. Branched-chain amino acid supplementation and indicators of muscle damage after endurance exercise. Int J Sport Nutr Exerc Metab. 2007;17(6):595-607.
  6. Huang WC, Chang YC, Chen YM, Hsu YJ, Huang CC, Kan NW, et al. Whey Protein Improves Marathon-Induced Injury and Exercise Performance in Elite Track Runners. Int J Med Sci. 2017;14(7):648-54.
  7. Valentine RJ, Saunders MJ, Todd MK, St Laurent TG. Influence of carbohydrate-protein beverage on cycling endurance and indices of muscle disruption. Int J Sport Nutr Exerc Metab. 2008;18(4):363-78.
  8. Betts JA, Toone RJ, Stokes KA, Thompson D. Systemic indices of skeletal muscle damage and recovery of muscle function after exercise: effect of combined carbohydrate-protein ingestion. Appl Physiol Nutr Metab. 2009;34(4):773-84.
  9. Etheridge T, Philp A, Watt PW. A single protein meal increases recovery of muscle function following an acute eccentric exercise bout. Appl Physiol Nutr Metab. 2008;33(3):483-8.
  10. Green MS, Corona BT, Doyle JA, Ingalls CP. Carbohydrate-protein drinks do not enhance recovery from exercise-induced muscle injury. Int J Sport Nutr Exerc Metab. 2008;18(1):1-18.
  11. Paddon-Jones D, Rasmussen BB. Dietary protein recommendations and the prevention of sarcopenia. Curr Opin Clin Nutr Metab Care. 2009;12(1):86-90.
  12. Witard OC, Wardle SL, Macnaughton LS, Hodgson AB, Tipton KD. Protein Considerations for Optimising Skeletal Muscle Mass in Healthy Young and Older Adults. Nutrients. 2016;8(4):181.
  13. Moore DR, Robinson MJ, Fry JL, Tang JE, Glover EI, Wilkinson SB, et al. Ingested protein dose response of muscle and albumin protein synthesis after resistance exercise in young men. Am J Clin Nutr. 2009;89(1):161-8.
  14. Trommelen J, van Loon LJ. Pre-Sleep Protein Ingestion to Improve the Skeletal Muscle Adaptive Response to Exercise Training. Nutrients. 2016;8(12).
  15. Berner LA, Becker G, Wise M, Doi J. Characterization of dietary protein among older adults in the United States: amount, animal sources, and meal patterns. J Acad Nutr Diet. 2013;113(6):809-15.
  16. Gillen JB, Trommelen J, Wardenaar FC, Brinkmans NY, Versteegen JJ, Jonvik KL, et al. Dietary Protein Intake and Distribution Patterns of Well-Trained Dutch Athletes. Int J Sport Nutr Exerc Metab. 2017;27(2):105-14.
  17. Moher D, Hopewell S, Schulz KF, Montori V, Gøtzsche PC, Devereaux PJ, et al. CONSORT 2010 Explanation and Elaboration: updated guidelines for reporting parallel group randomised trials. BMJ. 2010;340:c869.
  18. Koopman R, Pannemans DL, Jeukendrup AE, Gijsen AP, Senden JM, Halliday D, et al. Combined ingestion of protein and carbohydrate improves protein balance during ultra-endurance exercise. Am J Physiol Endocrinol Metab. 2004;287(4):E712-20.
  19. Koopman R, Wagenmakers AJ, Manders RJ, Zorenc AH, Senden JM, Gorselink M, et al. Combined ingestion of protein and free leucine with carbohydrate increases postexercise muscle protein synthesis in vivo in male subjects. Am J Physiol Endocrinol Metab. 2005;288(4):E645-53.
  20. Manninen AH. Hyperinsulinaemia, hyperaminoacidaemia and post-exercise muscle anabolism: the search for the optimal recovery drink. Br J Sports Med. 2006;40(11):900-5.
  21. Res PT, Groen B, Pennings B, Beelen M, Wallis GA, Gijsen AP, et al. Protein ingestion before sleep improves postexercise overnight recovery. Med Sci Sports Exerc [Internet]. 2012 Aug [cited 2021 Feb 19];44(8):1560–9. Available from: https://cris.maastrichtuniversity.nl/en/publications/protein-ingestion-before-sleep-improves-postexercise-overnight-re
  22. Kouw IWK, Holwerda AM, Trommelen J, Kramer IF, Bastiaanse J, Halson SL, et al. Protein ingestion before sleep increases overnight muscle protein synthesis rates in healthy older men: A randomized controlled trial. J Nutr [Internet]. 2017 Dec 1 [cited 2021 Feb 19];147(12):2252–61. Available from: https://pubmed.ncbi.nlm.nih.gov/28855419/
  23. Burd NA, Gorissen SH, van Vliet S, Snijders T, van Loon LJ. Differences in postprandial protein handling after beef com-pared with milk ingestion during postexercise recovery: a randomized controlled trial. Am J Clin Nutr. 2015 Oct;102(4):828-36. doi: 10.3945/ajcn.114.103184. Epub 2015 Sep 9. PMID: 26354539.
  24. Mamerow MM, Mettler JA, English KL, Casperson SL, Arentson-Lantz E, Sheffield-Moore M, et al. Dietary protein distribu-tion positively influences 24-h muscle protein synthesis in healthy adults. J Nutr [Internet]. 2014;144(6):876–80. Available from: https://www.ncbi.nlm.nih.gov/pubmed/24477298
  25. Areta JL, Burke LM, Ross ML, Camera DM, West DWD, Broad EM, et al. Timing and distribution of protein ingestion during prolonged recovery from resistance exercise alters myofibrillar protein synthesis. J Physiol. 2013;591(9):2319–31.
  26. Trommelen J, Kouw IWK, Holwerda AM, Snijders T, Halson SL, Rollo I, Verdijk LB, van Loon LJC. Presleep dietary protein-derived amino acids are incorporated in myofibrillar protein during postexercise overnight recovery. Am J Physiol Endo-crinol Metab. 2018 May 1;314(5):E457-E467. doi: 10.1152/ajpendo.00273.2016. Epub 2017 May 23. PMID: 28536184.
  27. Hoffman JR, Falvo MJ. Protein - Which is Best? J Sport Sci Med [Internet]. 2004;3(3):118–30. Available from: https://www.ncbi.nlm.nih.gov/pubmed/24482589

Not in article, only in rebuttal:

  1. Phillips SM, Tipton KD, Aarsland A, Wolf SE, Wolfe RR. Mixed muscle protein synthesis and breakdown after resistance exercise in humans. Am J Physiol [Internet]. 1997;273(1 Pt 1):E99-107. Available from: https://www.ncbi.nlm.nih.gov/pubmed/9252485
  2. Tipton KD, Ferrando AA, Phillips SM, Doyle D, Wolfe RR. Postexercise net protein synthesis in human muscle from orally administered amino acids. Am J Physiol - Endocrinol Metab. 1999;276(4 39-4).
  3. Gorissen SHM, Trommelen J, Kouw IWK, Holwerda AM, Pennings B, Groen BBL, et al. Protein type, protein dose, and age modulate dietary protein digestion and phenylalanine absorption kinetics and plasma phenylalanine availability in humans. J Nutr [Internet]. 2020 Aug 1 [cited 2021 Feb 23];150(8):2041–50. Available from: https://pubmed.ncbi.nlm.nih.gov/32069356/

Reviewer 2 Report

Introduction, par. 1, lines 2-3

It is known that repeated episodes of high intensity exercise diminish exercise induced muscle damage. So, the author should also refer to the repeated bout effect phenomenon which may have caused the differences that were observed among investigations regarding alterations in muscle damage biomarkers (Introduction section, par. 2).

Introduction, par. 1, lines 4-5

Delayed onset muscle soreness is an indirect biomarker of muscle damage. Please change this throughout the manuscript.

Methods, participants

Please provide the training status of the participants.

Methods, study design

Were the participants controlled for nutritional supplements during the race?

Methods, study design

The study design section is rather confusing. It would be clearer for the reader if procedures were divided in subsections pre-race, during race, post-race measurements.

The number of participants is also confusing (i.e., 419 screened but finally there were 2 groups of 75 and 74 each). More detailed/accurate presentation of the subjects is needed, not only in the text but also in figure 1. For instance, someone have το subtract the participants that were excluded, right?

Since in Table 1, there are demographics about men. The relevant information about women should also be included in Table 1.   

Methods, Measurements

Delayed onset muscle soreness is an indirect biomarker of muscle damage. Please change this throughout the manuscript.

Methods, Table 2

In this set of data, it is clear that there is no difference between the two groups regarding protein intake (i.e., g/kg/d). However, in the Results, Dietary intake, par. 2 the results are different, indicating differences between the two groups.

Table 4

The values are referring to measurements performed on day 1 or day 2 post road-race?

Figures 4 and 5

Please include values/description on axis x.

Discussion, par. 1

In reality, it was found that delayed onset muscle soreness was significantly larger after protein supplementation (Figure 3).

Discussion, par. 2

The authors should discuss the increase of protein intake (in g/kg/d) that was achieved after the protein supplementation.

Discussion, par. 4

In this paragraph, the authors have to discuss about the repeated bout effect phenomenon.

Author Response

February 24th 2021

We are pleased to submit the revised version of our research article entitled “The effect of protein supplementation versus carbohydrate supplementation on muscle damage markers and soreness following a 15-km road race: a double-blind randomized controlled trial” (nutrients-1095772).

We thank the reviewers for their thorough examination of the manuscript and revision and for their valuable comments and suggestions.

A point-by-point response to the comments from the reviewers, including revisions made to the manuscript in red, can be found below. Additionally, the revisions made to the manuscript are shown by track changes in the manuscript.

Reviewer 2

Introduction, par. 1, lines 2-3   
It is known that repeated episodes of high intensity exercise diminish exercise induced muscle damage. So, the author should also refer to the repeated bout effect phenomenon which may have caused the differences that were observed among investigations regarding alterations in muscle damage biomarkers (Introduction section, par. 2).          
We thank the reviewer for this good remark. We added this to the introduction, line 58-62.

‘’This discrepancy could be partly due to a difference in using a study protocol with or without multiple bouts of exercise, since the repeated bout effect could influence the findings. This repeated bout effect refers to the adaptation whereby a single bout of eccentric exercise protects against muscle damage from subsequent eccentric bouts (43).’’

Introduction, par. 1, lines 4-5
Delayed onset muscle soreness is an indirect biomarker of muscle damage. Please change this throughout the manuscript.
We adjusted this throughout the manuscript.

Methods, participants 
Please provide the training status of the participants. 
We have added the training status of the participants, quantified as number of training weeks, number of training sessions per week and the average distance per training session (Table 1 and line 269).

‘’There were no differences between the protein and placebo group for any of the baseline characteristics, including the training status of the participants (Table 1).’’

Methods, study design
Were the participants controlled for nutritional supplements during the race?
We asked the participants whether they normally use or around the race used any medication or supplements, including nutritional supplements. We clarified this in the methods section, line 201-202. We described in the results section, 3.7. Measures that could influence muscle soreness, line 377-387, that no differences between groups were observed in taken measures that could influence our study outcomes.

‘’We asked participants to report any treatment (e.g. muscle cream or having a massage or a bath), medications or other nutritional supplementation for pain relief or other reasons that may have interacted with our primary outcome.’’

The study design section is rather confusing. It would be clearer for the reader if procedures were divided in subsections pre-race, during race, post-race measurements.
We changed this according to the proposal of the reviewer, see line 115-138.

‘’2.2. Study design
In this double-blind, placebo-controlled intervention study a total of 323 eligible participants out of 419 screened participants (Figure 1) were randomly allocated to either the protein supplement or the CHO placebo supplement group. An independent researcher randomized the study participants by means of computer-generated random numbers with a block size of 10 in a 1:1 ratio.    
Pre-race: Baseline (i.e. 1 – 3 days pre-race) muscle soreness was questioned using online versions of the Short-Form Brief Pain Inventory (BPI-SF) (20), including a Numeric Pain Rating Scale (NPRS) (scale: 0-10). Moreover, habitual dietary intake was determined using 24-h recalls. Information about demographics (i.e. age, sex, height, body weight), exercise training characteristics were collected from all participants.

During race:
Due to the large number of participants in the race (±30,000), runners departed in 9 separate ‘waves’ between 13.00 and 14.00 CEST. Directly after finishing the race, participants received the first protein or placebo supplement to ingest immediately. Finish times were collected from all participants. Additionally, average heart rate during the race and exercise intensity was collected from participants that ran with a heart rate monitor.

Post-race: the participants received supplements to consume prior to sleep and during breakfast until three days post-race. Participants were instructed to fill out questionnaires about muscle soreness at 1 day, 2 days and 3 days post-race. Moreover, study participants completed 24-h recalls about their dietary intake on the day of the race, and the first and second day post-race.
Within a subgroup of 75 participants of the protein group and 74 participants of the placebo group, objective muscle soreness was measured with a strain gauge algometer at 25 – 48 h post-race. Blood concentrations of creatine kinase (CK) and lactate dehydrogenase (LDH) were also assessed in this subgroup to determine muscle damage. An overview of the study procedures is presented in Figure 2.’’

The number of participants is also confusing (i.e., 419 screened but finally there were 2 groups of 75 and 74 each). More detailed/accurate presentation of the subjects is needed, not only in the text but also in figure 1. For instance, someone have tο subtract the participants that were excluded, right?
We apologize for the confusion. The number of participants differed for our primary and secondary outcomes and for the Intention-To-Treat (ITT) and Per Protocol (PP) analysis. We had 323 participants in total, but not everyone filled in the questionnaire on each day. When a participant missed a questionnaire post-race, but filled out a questionnaire on another post-race day, they were included in the ITT analyses, but deleted for the PP analyses. The exact numbers for the ITT analyses of our subjectively perceived lower extremity muscle soreness are shown in Table 3. The sample size of the objectively measured pressure pain thresholds is given in Table 4 and we added the sample size of the muscle damage blood markers in the legend of Figure 5. For the PP analyses the numbers are given in the Supplemental Tables.

Our main goal of the study was to assess subjective measures of muscle soreness in a large group of participants, since most studies are only performed in small sample sizes. Furthermore, we wanted to confirm whether these outcomes correlated with objective measures of muscle soreness and muscle damage and invited study participants for this purpose for an additional study visit to our laboratory. Due to constraints in personnel and laboratory equipment (i.e. maximal number of blood drawings/day), this was only possible for a sub-group of participants. Nevertheless, this was only applicable to our secondary outcome measures and did not affect or primary research question.

We explained this further in the methods- study design and Figure 1 and described drop out reasons in the first paragraph of the results. Furthermore, we explained in the methods – statistical analyses, line 239-248, why we showed both the ITT and PP analysis, thus also resulting in different sample sizes as indicated in Figure 1.

‘’According to the CONSORT guidelines, primarily an intention-to-treat analysis (ITT) was performed and a per-protocol (PP) analysis was performed alongside to enable the influence of any missing data to be investigated (31). The PP analysis included per day the participants that filled out the questionnaire and reported to be perfectly compliant (i.e. 250 ml post-exercise, 250 ml with breakfast and 500 ml prior to sleep) until the analyzed day (one, two or three days post-race). Both analyses were performed to show both the results of the actual effect of the study protocol (PP analyses), but on the other hand also include the feasibility of a certain protocol in a real-life setting and the results in that setting (ITT analysis). ’’

Since in Table 1, there are demographics about men. The relevant information about women should also be included in Table 1.            
In Table 1 we describe the baseline characteristics of the total group (n=323) and we showed how many of these participants were male (181 i.e. 56%). The other 142 participants were female (44%). So data from both, men and women, were described in Table 1.

Methods, Measurements          
Delayed onset muscle soreness is an indirect biomarker of muscle damage. Please change this throughout the manuscript.      
We adjusted this throughout the manuscript.

Methods, Table 2           
In this set of data, it is clear that there is no difference between the two groups regarding protein intake (i.e., g/kg/d). However, in the Results, Dietary intake, par. 2 the results are different, indicating differences between the two groups.               
In Table 1 the habitual dietary intake has been described that was measured with two 24 hour recalls in the week prior to the race. We measured however also the intake on the day of the race and 1 and 2 days post-race. These data are shown in Table 2. We did this to account for possible differences in dietary intake on race days compared to general dietary intake.

Table 4
The values are referring to measurements performed on day 1 or day 2 post road-race?
Yes, this is correct. Approximately half of the participants arrived to our Research Centre on day 1 post race and the other half on day 2 post race for the measurements as explained in the methods section. We combined these datapoints, because we saw no time effect within our analyses (see below and line 350-351 within the results section).      

‘’The timing of the objective muscle soreness measurement was not significantly different between groups (P = 0.55).’’

Figures 4 and 5
Please include values/description on axis x.
We added the legends to figure 3, 4 and 5.

Discussion, par. 1
In reality, it was found that delayed onset muscle soreness was significantly larger after protein supplementation (Figure 3).
Thank you for noticing. We accidentally added the per protocol analyses pictures, while we meant to add the intention-to-treat analysis to this picture since that should be the primary analysis according to the CONSORT guidelines. We changed this now.

Discussion, par. 2
The authors should discuss the increase of protein intake (in g/kg/d) that was achieved after the protein supplementation.
We added this to the discussion, line 397-403.

‘’These findings indicate that 3 doses of post-exercise protein supplementation resulting in average protein intake of 1.94 ± 0.43 g/kg/d on race day, 1.97 ± 0.44 g/kg/d at 1 day post-race and 1.89 ± 0.40 g/kg/d at 2 days post-race are not more preferable than carbohydrate supplementation in the placebo group that consumed 1.10 ± 0.40 g/kg/d protein on race day, 1.09 ± 0.39 g/kg/d at 1 day post-race and 1.12 ± 0.34 g/kg/d at 2 days post-race to re-duce exercise-induced muscle soreness and other damage markers in recreational endurance athletes that already have a sufficient protein intake.’’

Discussion, par. 4
In this paragraph, the authors have to discuss about the repeated bout effect phenomenon.
We added this to the discussion line 436-442.

‘’Finally, the relatively low levels of exercise-induced muscle soreness and other muscle damage markers could contribute to the absent superior effects of protein supplementation. Peak levels of perceived muscle soreness were found one day post-race and were on average 2.96 ± 2.27 in the protein group and 2.46 ± 2.38 in the placebo group. These low muscle soreness values could be caused by the repeated bout effect phenomenon, that decreases muscle damage from subsequent eccentric bouts after one single bout (43), since our study participants train on average 2 times a week for 48 weeks a year.’’

Mentioned references in response to Reviewer 2

  1. USDA. “2015–2020 Dietary Guidelines for Americans, 8th edn” (2015).
  2. McHugh MP. Recent advances in the understanding of the repeated bout effect: The protective effect against muscle damage from a single bout of eccentric exercise [Internet]. Vol. 13, Scandinavian Journal of Medicine and Science in Sports. Scand J Med Sci Sports; 2003 [cited 2021 Feb 7]. p. 88–97. Available from: https://pubmed.ncbi.nlm.nih.gov/12641640/

Round 2

Reviewer 1 Report

The revised paper looks good now. For the repeated measures ANOVA, please report if the interaction effect (time x  treatment) was significant or not. This is more important than the main effect of treatment alone.

Author Response

Thank you. We clarified in the methods and results section that we reported the interaction effect.

Methods section, line 259-260

‘’A repeated measures ANOVA was performed to assess the interaction effect (time x treatment) on the subjectively measured muscle soreness for the PP analyses’’.

Results section, line 335-336

‘’We found no significant time*treatment interaction effect in a repeated measures ANOVA (p = 0.10).’’

Reviewer 2 Report

No further comments

Author Response

Thank you.